# Potential Causal Effects of Cystatin C on Age-Related Macular Degeneration: A Two-Sample Mendelian Randomization Study

**DOI:** 10.3390/biomedicines13112827

**Published:** 2025-11-20

**Authors:** Young Lee, Je Hyun Seo

**Affiliations:** 1Veterans Medical Research Institute, Veterans Health Service Medical Center, Seoul 05368, Republic of Korea; lyou7688@gmail.com; 2Department of Ophthalmology, Veterans Health Service Medical Center, Seoul 05368, Republic of Korea

**Keywords:** cystatin C, creatinine, age-related macular degeneration, Mendelian randomization, single-nucleotide polymorphisms

## Abstract

**Background/Objectives**: Previous studies have shown an association between kidney function and age-related macular degeneration (AMD). This study aims to assess whether the kidney function-related parameters of serum cystatin C and creatinine levels are associated with increased risk of AMD and its subtypes. **Methods**: Genetic instruments for variants associated with serum cystatin C and creatinine levels as exposure at genome-wide significance (*p* < 5.0 × 10^−8^) were obtained from the UK Biobank. Genetic data for AMD and its subtypes were obtained from the FinnGen project. A two-sample Mendelian randomization analysis was performed to evaluate the causal effects of serum cystatin C and creatinine levels on AMD and its subtypes. **Results**: Using an inverse-variance weighted approach, higher cystatin C levels are associated with an increased risk of AMD [odds ratio (OR) = 1.13, 95% confidence interval (CI): 1.04 to 1.22, *p* = 0.004 for overall AMD; OR = 1.14, 95% CI: 1.04 to 1.25, *p* = 0.007 for dry AMD; OR = 1.14, 95% CI: 1.03 to 1.26, *p* = 0.011 for wet AMD]. However, serum creatinine levels did not significantly impact the risk of AMD or its subtypes. **Conclusions**: This study provides genetic evidence that higher cystatin C levels may be a causal risk factor for AMD and its subtypes, whereas serum creatinine was not. This result implies the need to investigate the effect of cystatin C on AMD potentially independent of kidney function.

## 1. Introduction

Age-related macular degeneration (AMD) is a degenerative eye condition that progressively impacts the macula, the central area of the retina crucial for sharp and detailed vision [1]. AMD includes dry AMD (early) and wet AMD (late); dry AMD is characterized by drusen accumulation and pigmentary changes [2,3], often without noticeable symptoms; however, it can progress to more severe, wet AMD characterized by two subtypes, i.e., geographic atrophy (GA) and choroidal neovascularization (CNV), that lead to serious vision impairment [4,5]. The prevalence of AMD is projected to increase by 47% to nearly 288 million individuals worldwide in the next 20 years, posing a major burden to health care systems across the world [6]. According to a study by Wong et al., the global prevalence of all types of AMD is approximately 8.7% worldwide [7]. In Europe, the prevalence of dry AMD increases from 3.5% in those aged 55–59 years to 17.9% in those aged 85 years; for wet AMD, the prevalence increases from 0.1% to 9.8% [8]. In Asia, the prevalence rates of dry and wet AMD were found to range from 1.4% to 17.3% and 0.1% to 7.3%, respectively, with a higher prevalence in elderly groups [9,10]. Observational epidemiological studies have reported associations of lifestyle and metabolic risk factors such as smoking, alcohol intake, obesity, glycemic traits, and dyslipidemia with AMD [11]. The pathogenesis of AMD is linked to chronic inflammation, lipid deposition, oxidative stress, and breakdown of the extracellular matrix [12]. However, these findings are inconsistent and cannot be established as causal owing to limitations introduced by confounding and reverse causality. Despite significant advances in clinical care and a deeper understanding of the risk factors of AMD, considerable challenges persist in fully addressing the medical needs of those affected by this condition.

Cystatin C is a cysteine protease inhibitor that is produced and released constantly by all nucleated cells [13]. As cystatin C is freely filtered at the glomerulus and metabolized in the proximal tubule, it is a sensitive marker of kidney function, especially in the early course of disease [14]. To date, the connection between serum cystatin C levels and AMD is not well understood. According to the Beaver Dam Eye study, serum cystatin C level at baseline was associated with the incidence of dry AMD (odds ratio [OR]: 1.16; 95% confidence interval [CI] 1.16–1.35) and exudative AMD (OR: 1.42, 95% CI:1.03–1.96) [15]. Because cystatin C is associated with renal function, it has been hypothesized that a decline in renal function may be related to AMD. Subsequent research in the Multi-Ethnic Study of Atherosclerosis study, however, failed to find a connection between dry AMD and renal function [16]. A recent study by the Asian Eye Epidemiology Consortium demonstrated that chronic kidney disease (CKD) and estimated glomerular filtration rate (eGFR) were not associated with dry AMD (all *p* ≥ 0.149), whereas CKD was associated with wet AMD (OR: 1.46, 95% CI: 1.11–1.93, *p* = 0.008) [17]. In addition, cystatin C has been studied for its extra-renal roles, including its involvement in inflammation regulation, and neurological functions [18,19,20]. In addition, since cystatin C is expressed in ocular tissues [21,22,23], analyzing its association with AMD pathogenesis would be of significant research value.

Mendelian randomization (MR) uses genetic variation as an instrumental variable (IV) to identify potential causal links between risk factors and diseases [24,25,26]. Recent MR studies have been applied to clarify the causal relationships between systemic biomarkers (lifestyle, inflammation and metabolic factors) and AMD risk, offering insights beyond observational limitations [27,28,29,30]. These findings underscore the utility of MR in identifying potentially modifiable risk factors for AMD. This technique is based on the idea that genetic variants occur randomly at conception and are unaffected by conventional confounders. Because of this random allocation, MR analysis is less prone to confounding and reverse causality biases than traditional observational studies. In our study, we used a two-sample MR approach to investigate the causal relationships of the renal function markers cystatin C and creatinine with AMD and its subtypes (dry AMD, wet AMD) in a European population.

## 2. Materials and Methods

### 2.1. Study Design

The Institutional Review Board of the Veterans Health Service Medical Center approved this study protocol and informed consent waiver (IRB No. 2023-12-030; 11 January 2024) since the study was performed in a retrospective manner and utilized anonymized data. In addition, the study was conducted in compliance with the Helsinki Declaration.

### 2.2. Data Sources

Figure 1 illustrates the schematic diagrams of the analytical study design. We obtained all datasets from a European population. The genome-wide association study (GWAS) summary statistics datasets used in this analysis for exposure were obtained from the Pan-UK Biobank (https://pan.ukbb.broadinstitute.org/downloads/index.html, accessed on 17 June 2024), and those for outcome assessment were from FinnGen (https://finngen.gitbook.io/documentation/data-download, accessed on 4 November 2023) R9. In this MR analysis, AMD (*n* = 357,849, with 8913 cases and 348,936 controls) was examined as the outcome, with cystatin C (*n* = 400,940) identified as the primary exposure variable. Subtypes of AMD, namely dry AMD (*n* = 257,107, with 6065 cases and 251,042 controls) and wet AMD (*n* = 257,125, with 4848 cases and 252,277 controls), were also considered as outcomes. Additionally, serum creatinine (*n* = 400,761), a marker of kidney function, was included in the MR analysis to assess its potential association with AMD. In the Pan-UK Biobank, cystatin C (Field ID: 30720, unit: mg/L) and serum creatinine (Field ID: 30700, unit: µmol/L) were measured from serum samples. Both traits were inverse-rank normal transformed prior to genome-wide association analysis to reduce skewness and ensure comparability across participants. The datasets used for summary statistics are detailed in Table 1.

### 2.3. Selection of the Genetic Instrumental Variables

SNPs associated with exposure at a genome-wide significance level (*p* < 5.0 × 10^−8^) were utilized as IVs. To ensure independence among IVs, these SNPs underwent pruning for linkage disequilibrium (LD; *r*^2^ < 0.001, clumping distance = 10,000 kb). The LD calculations between SNPs were based on the 1000 Genomes Project Phase III European dataset. All palindromic SNPs (A/T or G/C) were excluded before LD clumping to avoid ambiguity in strand alignment. SNPs absent from either the exposure or outcome GWAS datasets were also removed to maintain consistency. The exposure and outcome datasets were harmonized using the harmonise_data function in the TwoSampleMR (version 0.5.6) R package (version 3.6.3) to align effect alleles, correct strand orientation, and ensure that all beta coefficients corresponded to the same effect allele across datasets. The *F* statistics, calculated to identify any potential issues with weak instruments, indicated that values greater than 10 suggest the absence of weak instrument bias [31].

### 2.4. Mendelian Randomization

When conducting MR analysis, it is crucial to adhere to several foundational assumptions: (1) the selected genetic variants are robustly associated with the exposure; (2) the variants are not associated with confounding factors; and (3) their effects on the outcome occur exclusively through the exposure, without horizontal pleiotropy. The primary analytical method was inverse variance-weighted (IVW) MR with multiplicative random effects, which provides reliable estimates under the assumption that all genetic instruments are valid [31,32,33]. To ensure robustness, we complemented IVW with additional sensitivity analyses, including the weighted median approach [34] and MR-Egger regression (applied with or without adjustment via the Simulation Extrapolation [SIMEX] method) [35]. The weighted median method can yield consistent estimates even when up to 50% of the instruments are invalid [34]. The MR-Egger regression allows detection and correction of unbalanced horizontal pleiotropy by estimating a non-zero intercept, while the SIMEX approach further adjusts for bias when the No Measurement Error (NOME) assumption is violated, particularly when *I*^2^ is less than 90% [36]. We assessed heterogeneity using Cochran’s Q statistic for IVW and Rücker’s Q′ statistic for MR-Egger. Evidence of heterogeneity may suggest the presence of pleiotropic variants [32,37]. In addition, MR pleiotropy residual sum and outlier (MR-PRESSO) analysis was conducted to identify and correct for horizontal pleiotropic outliers, improving the accuracy of causal estimates [35]. The distortion test implemented in MR-PRESSO was used to assess whether the causal estimates before and after outlier removal differed significantly. All MR analyses were performed using the TwoSampleMR (version 0.5.6) and simex (version 1.8) package in R version 3.6.3 (R Core Team, Vienna, Austria). Statistical power for each exposure–outcome association was estimated using the mRnd online tool (https://shiny.cnsgenomics.com/mRnd/, accessed on 3 November 2025). To further validate that the observed associations were not driven by distinct causal variants in linkage disequilibrium (LD), we performed locus-specific colocalization analyses (±500 kb around each lead variant) using the coloc.abf function (priors p1 = 1 × 10^−4^, p2 = 1 × 10^−4^, p12 = 1 × 10^−5^). The posterior probabilities were interpreted as follows: PP.H0—no association with either trait; PP.H1—association with exposure only; PP.H2—association with outcome only; PP.H3—both traits associated but with different causal variants; and PP.H4—both traits share the same causal variant. A PP.H4 > 0.8 was considered supportive of a shared causal signal between cystatin C and AMD. To assess the robustness of MR estimates, we repeated all MR analyses after excluding PP.H3-dominant loci (PP.H3 > 0.5), which may indicate distinct linked variants within the same genomic region.

## 3. Results

### 3.1. Selection of Instrumental Variables

Numbers of IVs were 446 and 416 for cystatin C and creatinine, respectively. The mean *F* statistics for IVs were 89.92 for cystatin C and 92.83 for serum creatinine (Table 2). All of the *F* statistics for IV MR were greater than 10, indicating a low chance of weak instrument bias. Appendix A provides detailed information about the IVs used in this study.

### 3.2. Heterogeneity and Horizontal Pleiotropy of the Instrumental Variables

We evaluated the NOME assumption using *I*^2^ and assessed heterogeneity through Cochran’s Q test and Rücker’s Q’ test, while also conducting analyses for horizontal pleiotropy. In all cases, the NOME assumption was satisfied (*I*^2^ value > 90), as detailed in Table 2. Heterogeneity was detected in all cases (Cochran’s Q, all *p* < 0.05; Rücker’s Q’, all *p* < 0.05, see Table 2), but the MR-Egger regression intercepts revealed no evidence of horizontal pleiotropic effects (all *p* > 0.05), with or without SIMEX adjustment. The MR-PRESSO global test was significant for all analyzed cases, and IVs identified as outliers due to horizontal pleiotropic effects are listed in Appendix A. Based on these findings, we recommend the use of MR-PRESSO for all cases [38].

### 3.3. Mendelian Randomization

Figure 2 and Figure 3 show forest plots of MR analyses for cystatin C and serum creatinine, respectively. These analyses explore their associations with outcomes of AMD, dry AMD, and wet AMD. Figure 4 depicts MR results using scatter plots for both cystatin C and serum creatinine, where the slope of the regression line represents the estimated causal effect size.

Figure 2 illustrates the analysis of cystatin C as an exposure and its associations with AMD and its subtypes. The IVW method demonstrated a significant association, indicating that higher cystatin C levels are associated with increased risk of AMD (OR = 1.13, 95% CI: 1.04 to 1.22, *p* = 0.004 for overall AMD; OR = 1.14, 95% CI: 1.04 to 1.25, *p* = 0.007 for dry AMD; OR = 1.14, 95% CI: 1.03 to 1.26, *p* = 0.011 for wet AMD). Although the weighted median and MR-Egger methods did not reach statistical significance, the effect estimates were directionally consistent with the IVW results (Figure 2). Confirmatory results were provided by the MR-PRESSO method, also included in Figure 2, which also supported the significant findings of the IVW method, particularly for overall AMD (OR = 1.12, 95% CI: 1.04 to 1.22, *p* = 0.004), dry AMD (OR = 1.12, 95% CI: 1.02 to 1.22, *p* = 0.013), and wet AMD (OR = 1.11, 95% CI: 1.01 to 1.22, *p* = 0.025). The MR-PRESSO distortion test indicated no significant difference in causal estimates before and after outlier correction (*p* = 0.852 for AMD, 0.714 for dry AMD, and 0.581 for wet AMD), confirming that outlier removal did not materially influence the overall results. Our comprehensive MR analysis consistently demonstrated that serum creatinine levels (Figure 3) did not significantly impact the risk of AMD or its subtypes. The direction of the observed associations did not indicate a consistent increase or decrease in AMD risk associated with changes in creatinine levels across all evaluated MR methods of IVW, weighted median, MR-Egger, and MR-PRESSO. The MR-PRESSO distortion test likewise showed no significant change in causal estimates after outlier correction (*p* = 0.317 for AMD, 0.838 for dry AMD, and 0.108 for wet AMD). In the IVW analyses, genetically predicted cystatin C had high power for AMD (97%) and dry AMD (93%), and still adequate power for wet AMD (86%). When power was recalculated using the MR-PRESSO estimates, power for cystatin C remained high for AMD (94%) and dry AMD (83%), but decreased for wet AMD to 67%. By contrast, analyses using serum creatinine as the exposure were underpowered (14% for AMD, 39% for dry AMD, and 6% for wet AMD; 6–30% with MR-PRESSO), so the non-significant creatinine results should be interpreted cautiously (Appendix A).

We also conducted locus-specific colocalization across 446 cystatin C instrument regions (Appendix A). For AMD, two loci showed strong evidence for a shared causal variant (PP.H4 ≈ 0.99). For the subtypes, one locus each for dry AMD and wet AMD showed strong colocalization (PP.H4 ≈ 0.99). Most regions were PP.H1-dominant, indicating limited local signal, whereas a subset were PP.H3-dominant (PP.H3 > 0.5), consistent with distinct linked variants within the same loci. Excluding PP.H3-dominant loci yielded materially unchanged MR estimates (Appendix A).

## 4. Discussion

In this study, we investigated the potential causal relationships between kidney function-related parameters and AMD using an MR approach. Our results provide genetic evidence that elevated serum cystatin C levels are associated with increased risk of AMD and its subtypes, whereas serum creatinine levels did not demonstrate a significant effect. These findings suggest that cystatin C may serve as a potential biomarker for identifying individuals at increased risk of AMD, beyond its conventional role as a marker of kidney function. Incorporating cystatin C measurements into clinical screening frameworks could improve early detection and refine risk stratification for both dry and wet AMD. Moreover, elucidating the molecular pathways linking cystatin C to retinal degeneration may open new therapeutic avenues, providing opportunities for targeted interventions aimed at modulating cystatin C-related mechanisms in AMD pathogenesis.

Several observational studies have shown that CKD increased the risk of AMD [39,40,41]. In the Beaver Dam Eye Study, baseline serum cystatin C was associated with increased incidence of dry AMD (OR, 1.16; 95% CI, 1.01–1.35) and exudative AMD (OR, 1.42; 95% CI, 1.03–1.94) after adjusting for age and other risk factors [15]. As serum cystatin C is widely used as a marker of kidney function for research [42], these findings raise questions about the potential association between kidney function and AMD. In a subsequent population-based cross-sectional study, high serum cystatin C levels and low eGFR (≤60 mL/min/1.73 m^2^) were not associated with dry AMD [16]. However, in a recent large multicenter study including 51,253 participants across 10 Asian populations, CKD was significantly associated with wet AMD. Nevertheless, CKD and eGFR were not significantly associated with dry AMD [17]. However, not all studies support an association between CKD and AMD. A recent MR study on the association between CKD using eGFR and AMD found no evidence of a causal relationship [43], which is consistent with our findings showing that serum creatinine levels did not have a significant effect on AMD. A cross-sectional study involving 5874 participants indicated no association between impaired kidney function and dry AMD [16]. In addition, the Singapore Epidemiology of Eye Diseases study revealed no link between CKD and AMD in adults [44]. Moreover, a multimodal retinal imaging study found no association between renal function parameters and AMD features, even after adjusting for age [45]. Previous observational studies reporting null associations may have been confounded by residual differences in kidney function, reverse causality, or measurement variability in cystatin C assays. In contrast, MR analyses, by leveraging genetic instruments, are less susceptible to such biases. This study may suggest that the cellular and pathophysiological functions of cystatin C deserve closer investigation beyond its conventional role as a marker of kidney function.

Cystatin C is an inhibitor of cysteine proteases and has been implicated in extracellular matrix remodeling, angiogenesis, and inflammatory pathways [46]. Experimental evidence has also suggested that dysregulation of cystatin C may contribute to abnormal protein aggregation and complement activation, processes known to play a central role in AMD development [21]. Previous studies have shown that cystatin C is abundantly expressed and directionally secreted from the retinal pigment epithelium (RPE) [47]. Studies in cell cultures and animals suggest that cathepsins support retinal photoreceptor and Bruch’s membrane health, including by releasing antiangiogenic endostatins from Bruch’s membrane collagen [48]. The harmful effect of elevated serum cystatin C is thought to arise from its interference with the protective role of cathepsins in releasing anti-angiogenic endostatins from Bruch’s membrane collagen, increasing the risk of exudative AMD. A recent study has shown that variant B cystatin C, resulting from the A25T mutation, is mistrafficked with partial mitochondrial retention in RPE cells, interacts specifically with mitochondrial outer membrane proteins, and increases susceptibility to mitochondrial ROS damage, which may be linked to the pathogenesis of AMD [49,50]. Given that cystatin C is a potent endogenous inhibitor of lysosomal cathepsins, we considered the possibility of horizontal pleiotropy mediated via cathepsin-axis genes (e.g., *CTSB*, *CTSD*, *CTSK*, *CTSS*, and *CTSV*). In this MR study, SNP (rs73102387, *CST3*; *CST4*), SNPs (rs78516764, rs111701119, rs118011290) located in an intergenic *CST5*; *GGTLC1* and rs7688550 (*CTSO*; *PDGFC*) were used as instrumental variables, which may have influenced the analytical results. Genetic variants in these genes could influence AMD risk through pathways independent of cystatin C and renal function, for instance via extracellular matrix remodeling, angiogenesis, or choroidal neovascularisation. Accordingly, while our MR framework mitigates many confounding influences, residual pleiotropic effects via cathepsin-related genes cannot be fully excluded.

The main strength of our study lies in the use of a large cohort dataset, which allowed us to explore the potential causal relationships of serum cystatin C and creatinine with AMD and dry/wet AMD. Our findings have several important implications. First, cystatin C may serve as a novel biomarker for AMD risk prediction, independent of kidney function. Elucidating the molecular pathways linking cystatin C to retinal degeneration could provide new therapeutic targets for AMD prevention and treatment. The null finding observed with creatinine should be interpreted with caution, as cystatin C is less affected by muscle mass and serves as a more sensitive biomarker for detecting early reductions in eGFR compared with creatinine [51]. Second, these results underscore the importance of distinguishing between kidney function markers when investigating systemic risk factors for ocular diseases. However, this study also has several limitations. First, the genetic instruments were derived from individuals of predominantly European ancestry, which may limit the generalizability of our findings to other populations. Second, although MR reduces residual confounding, horizontal pleiotropy cannot be entirely excluded. The IVs used in this analysis might still influence AMD risk through undetected pleiotropic pathways. Third, limitations related to data sources and study design should be considered. Both the exposure and outcome GWAS datasets were obtained from volunteer-based biobank cohorts, which could introduce selection bias (commonly referred to as healthy volunteer bias) and may affect the representativeness of renal and ophthalmic trait associations. Furthermore, as this was a two-sample MR study based solely on summary-level GWAS data, it was not possible to perform non-linear or piecewise MR analyses that require individual-level data to construct quantile-specific genetic instruments. Similarly, direct cohort-level or experimental validation could not be conducted within the current framework. Future MR studies using individual-level data and independent cohorts will be essential to assess potential nonlinearity and to confirm the robustness of these causal relationships. Finally, statistical power differed substantially between exposures. For cystatin C, power was high for AMD (97%) and dry AMD (93%), and acceptable for wet AMD based on IVW estimates (86%). However, when MR-PRESSO estimates were used, power for wet AMD was lower (67%). Moreover, creatinine-based analyses had clearly insufficient power (14% for AMD, 39% for dry AMD, and 6% for wet AMD based on IVW; 6–30% with MR-PRESSO). Accordingly, the null findings for creatinine should be interpreted with caution, as they may reflect insufficient power rather than the absence of a true association. Future research should validate these associations in diverse cohorts, explore the mechanistic basis of cystatin C in AMD development, and investigate potential interactions with environmental risk factors such as smoking or diet.

In conclusion, our study provides evidence that genetically elevated cystatin C levels are causally associated with increased risk of AMD and its subtypes, whereas serum creatinine is not. These findings may be partly explained by the greater sensitivity of cystatin C compared with creatinine, highlighting the need for further investigation into the role of cystatin C in retinal disease pathogenesis and raising the possibility of novel avenues for risk stratification and therapeutic intervention.

## Figures and Tables

**Figure 1 biomedicines-13-02827-f001:**
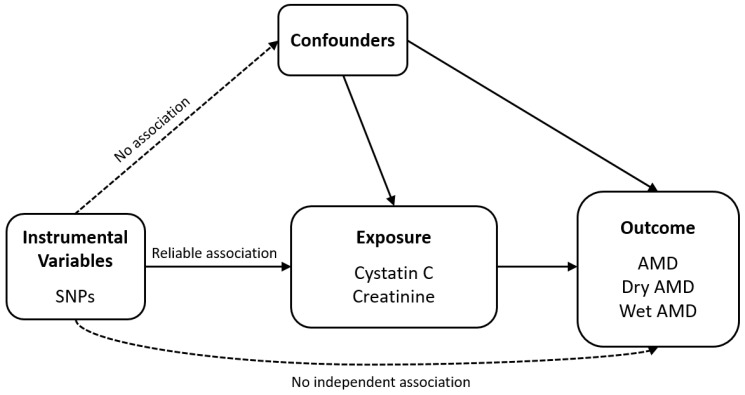
Schematic of the analytical study design. Abbreviations: AMD, age-related macular degeneration; SNP, single-nucleotide polymorphism.

**Figure 2 biomedicines-13-02827-f002:**
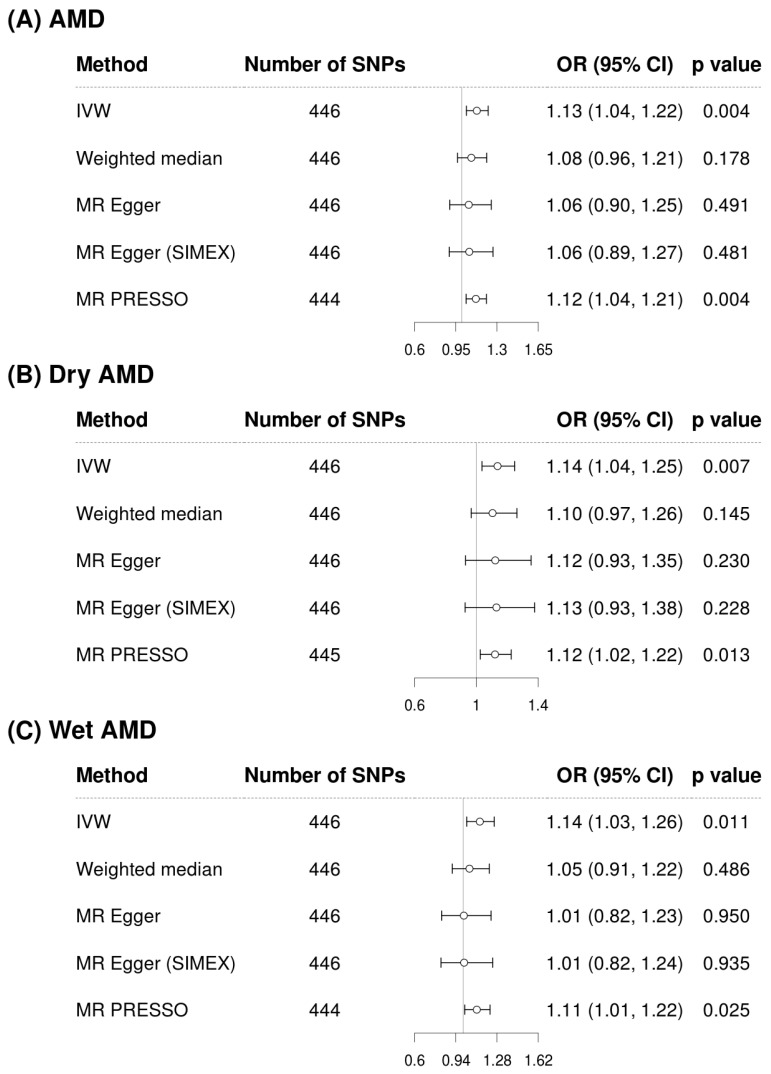
Forest plots of causal associations of cystatin C with AMD, dry AMD, and wet AMD. Abbreviations: AMD, age-related macular degeneration; CI, confidence interval; IVW, inverse-variance weighted; MR, Mendelian randomization; OR, odds ratio; PRESSO, pleiotropy residual sum and outlier; SIMEX, simulation extrapolation; SNP, single-nucleotide polymorphism.

**Figure 3 biomedicines-13-02827-f003:**
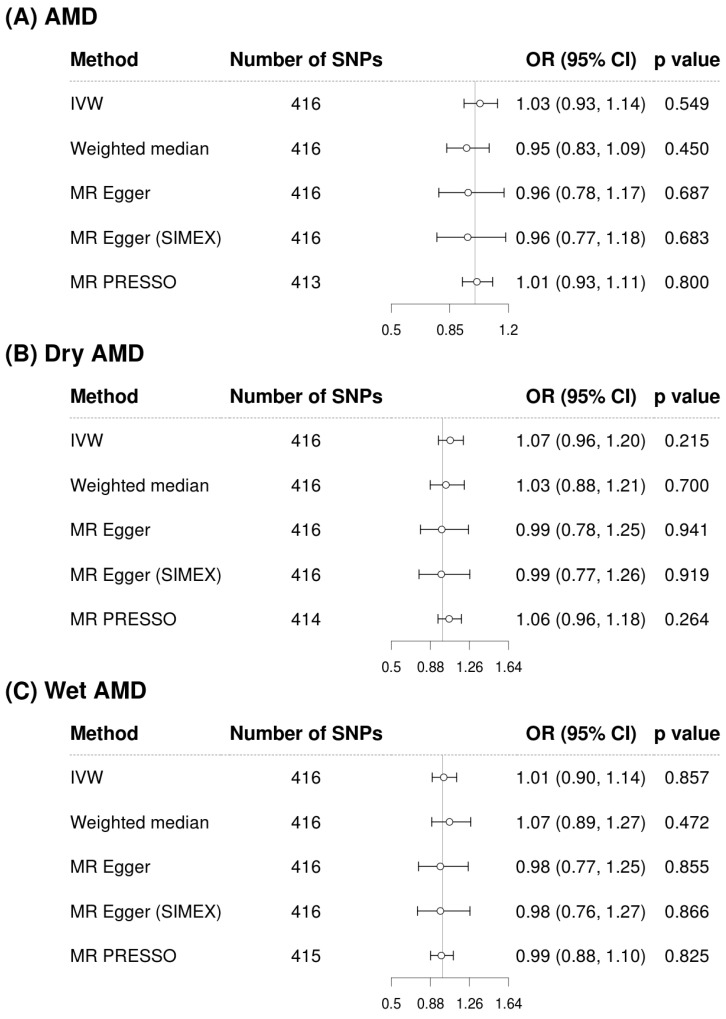
Forest plots of causal associations of creatinine with AMD, dry AMD, and wet AMD. Abbreviations: AMD, age-related macular degeneration; CI, confidence interval; IVW, inverse-variance weighted; MR, Mendelian randomization; OR, odds ratio; PRESSO, pleiotropy residual sum and outlier; SIMEX, simulation extrapolation; SNP, single-nucleotide polymorphism.

**Figure 4 biomedicines-13-02827-f004:**
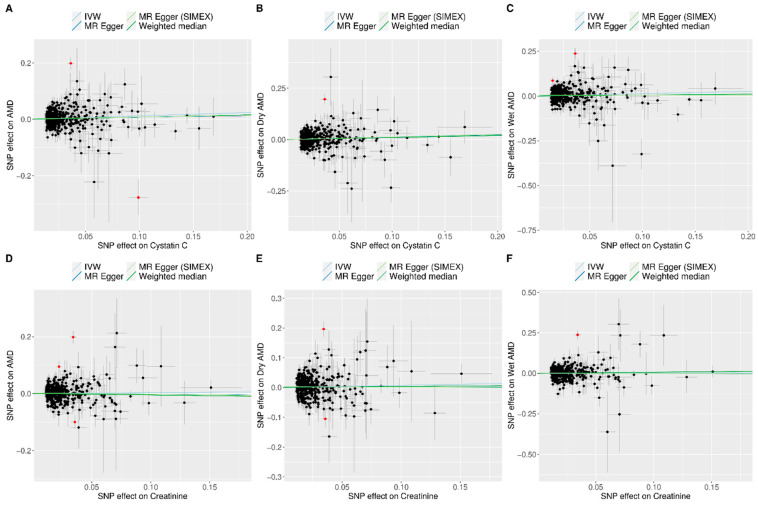
Scatter plots of MR tests assessing the effects of cystatin C or creatinine levels on AMD, dry AMD, and wet AMD. Scatter plot indicating the effects of cystatin C on AMD (**A**), dry AMD (**B**), and wet AMD (**C**) as well as those of creatinine on AMD (**D**), dry AMD (**E**), and wet AMD (**F**). Light blue, dark blue, light green, and dark green regression lines represent the IVW, MR-Egger, MR-Egger (SIMEX), and weighted median estimates, respectively. The slope of the line represents the causal effect of each method. Each dot corresponds to a SNP, with the x-axis representing the association between the SNP and the exposure, and the y-axis representing the association between the SNP and the outcome. Red dots indicate outliers in the MR-PRESSO analysis. Abbreviations: AMD, age-related macular degeneration; IVW, inverse-variance weighted; MR, Mendelian randomization; PRESSO, pleiotropy residual sum and outlier; SIMEX, simulation extrapolation; SNP, single-nucleotide polymorphism.

**Table 1 biomedicines-13-02827-t001:** Summary statistics of data sources.

Traits	Data Source	No. of Participants	Population	No. of Variants	URL
Cystatin C	UKB	400,940	European	23,012,599	https://pan.ukbb.broadinstitute.org/downloads/index.html
Creatinine	UKB	400,761	European	23,011,696	
AMD	FinnGen	357,849 (8913 cases + 348,936 controls)	European	20,169,869	https://finngen.gitbook.io/documentation/data-download
Dry AMD	FinnGen	257,107 (6065 cases + 251,042 controls)	European	20,165,949	
Wet AMD	FinnGen	257,125 (4848 cases + 252,277 controls)	European	20,165,938	

Abbreviations: AMD, age-related macular degeneration; UKB, UK Biobank.

**Table 2 biomedicines-13-02827-t002:** Heterogeneity and horizontal pleiotropy of instrumental variables.

Exposure	Outcome				Heterogeneity		Horizontal Pleiotropy				
								MR-Egger		MR-Egger (SIMEX)	
		N	F	*I*^2^ (%)	*p* *	*p †*	*p ‡*	Intercept, β (SE)	*p*	Intercept, β (SE)	*p*
Cystatin C	AMD	446	89.92	95.20	<0.001	<0.001	<0.001	0.002 (0.002)	0.395	0.002 (0.002)	0.435
Creatinine		416	92.83	94.61	<0.001	<0.001	<0.001	0.002 (0.003)	0.431	0.002 (0.003)	0.436
Cystatin C	Dry AMD	446	89.92	95.20	<0.001	<0.001	<0.001	0.000 (0.003)	0.876	0.000 (0.003)	0.926
Creatinine		416	92.83	94.61	<0.001	<0.001	<0.001	0.002 (0.003)	0.444	0.002 (0.003)	0.447
Cystatin C	Wet AMD	446	89.92	95.19	<0.001	<0.001	<0.001	0.004 (0.003)	0.169	0.004 (0.003)	0.189
Creatinine		416	92.83	94.61	<0.001	<0.001	<0.001	0.001 (0.003)	0.759	0.001 (0.003)	0.771

*p* *: Cochran’s Q test from IVW; *p †*: Rücker’s Q’ test from MR-Egger; *p ‡*: MR-PRESSO global test. Abbreviations: β, beta coefficient; AMD, age-related macular degeneration; F, mean F statistic; IVW, inverse-variance weighted; MR, Mendelian randomization; N, number of instruments; PRESSO, pleiotropy sum of residuals and outlier; SE, standard error; SIMEX, simulation extrapolation.

## Data Availability

The datasets used and/or analyzed in the current study are available from the Pan-UK Biobank (https://pan.ukbb.broadinstitute.org/downloads/index.html, accessed on 17 June 2024) and FinnGen (https://finngen.gitbook.io/documentation/data-download, accessed on 4 November 2023). The code for the main analyses in this study is available on GitHub (https://github.com/lyou7688/cystatinC_AMD_MR, commit ee1e4df, accessed on 3 November 2025).

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
