# Peer review of "Potential Causal Effects of Cystatin C on Age-Related Macular Degeneration: A Two-Sample Mendelian Randomization Study"

_biomedicines, 2025, doi:10.3390/biomedicines13112827_

Round 1

Reviewer 1 Report

Comments and Suggestions for Authors

This manuscript investigates the potential causal relationship between serum cystatin C and age-related macular degeneration using a two-sample Mendelian randomization. The topic is clinically relevant and interesting. However, while the results are statistically significant, several aspects of the study require substantial revision.

1. This manuscript consists solely of bioinformatics analysis of public databases which are not accompanied by validation. Author should do cohort study to support MR findings.Minor

2. grammatical issues are present (e.g., "odd ratio" → "odds ratio").

3. In the introduction section, author should provide a detailed introduction to the specific applications of Mendelian randomization in the macular degeneration field and cite PMID: 39482668.

4. Reference [17] is misdated ("update for summer 2023" should be "update 2023, Wellcome Open Research"). Check all references for formatting consistency.

5. Data and code need to be shared either through a code-sharing repo like GitHub or a docker-like system such as codeocean for clear reproducibility of the work. 

6. Provide a sample size calculation and statistical power analysis to assess whether the current sample size is sufficient to support the study's conclusions.

7. A conclusion figure (graphical abstract) will be very useful for the readers.

Author Response

This manuscript investigates the potential causal relationship between serum cystatin C and age-related macular degeneration using a two-sample Mendelian randomization. The topic is clinically relevant and interesting. However, while the results are statistically significant, several aspects of the study require substantial revision.

  1. This manuscript consists solely of bioinformatics analysis of public databases which are not accompanied by validation. Author should do cohort study to support MR findings.Minor

 → We appreciate this valuable comment. As our study was designed as a hypothesis-generating two-sample MR analysis using publicly available GWAS summary statistics, direct validation using independent cohort-level data could not be performed within the current scope. Nevertheless, we fully agree that external validation would strengthen the causal inference and enhance clinical relevance. To address this concern, we have explicitly acknowledged the absence of cohort-level or experimental validation as a key limitation in the Discussion section (lines 318-328) and emphasized that future MR studies with individual-level or prospective cohort data are warranted to confirm and extend these findings.

“Third, limitations related to data sources and study design should be considered. Both the exposure and outcome GWAS datasets were obtained from volunteer-based biobank cohorts, which could introduce selection bias (commonly referred to as healthy volunteer bias) and may affect the representativeness of renal and ophthalmic trait associations. Furthermore, as this was a two-sample MR study based solely on summary-level GWAS data, it was not possible to perform non-linear or piecewise MR analyses that require individual-level data to construct quantile-specific genetic instruments. Similarly, direct cohort-level or experimental validation could not be con-ducted within the current framework. Future MR studies using individual-level data and independent cohorts will be essential to assess potential nonlinearity and to con-firm the robustness of these causal relationships.”

  1. grammatical issues are present (e.g., "odd ratio" → "odds ratio").

→ Thank you for your comments. We carefully reviewed and corrected all typographical and grammatical errors throughout the manuscript, including replacing “odd ratio” with “odds ratio”.

  1. In the introduction section, author should provide a detailed introduction to the specific applications of Mendelian randomization in the macular degeneration field and cite PMID: 39482668.

→ Thank you for the helpful suggestion. We have expanded the Introduction to include a more detailed discussion of MR applications in AMD research and cited the recommended reference (PMID: 39482668):

(lines 72-76)

“Recent MR studies have been applied to clarify the causal relationships between systemic biomarkers (lifestyle, inflammation and metabolic factors) and AMD risk, offering insights beyond observational limitations [27-30]. These findings underscore the utility of MR in identifying potentially modifiable risk factors for AMD.”

  1. Reference [17] is misdated ("update for summer 2023" should be "update 2023, Wellcome Open Research"). Check all references for formatting consistency.

→ Thank you for your comments. Due to a grammatical error in the reference, I searched again, but confirmed that the name was correct. Thank you for careful comments:

Wellcome Open Res. 2023 Aug 4:4:186. doi: 10.12688/wellcomeopenres.15555.3. eCollection 2019. Guidelines for performing Mendelian randomization investigations: update for summer 2023

  1. Data and code need to be shared either through a code-sharing repo like GitHub or a docker-like system such as codeocean for clear reproducibility of the work. 

→ We appreciate this important comment. All analyses in this study were conducted using publicly available GWAS summary statistics from the Pan-UK Biobank and FinnGen. All code used for the main analyses has been made publicly available on GitHub to ensure transparency and reproducibility:

“The code for the main analyses in this study is available on GitHub (https://github.com/lyou7688/cystatinC_AMD_MR).”

We have included this statement in the Data Availability section.

  1. Provide a sample size calculation and statistical power analysis to assess whether the current sample size is sufficient to support the study's conclusions.

→ We thank the reviewer for this important comment. We have now performed statistical power calculations using the mRnd power calculator (http://shiny.cnsgenomics.com/mRnd/), based on the proportion of variance explained by the instruments, the sample sizes of the outcome GWAS, and the IVW and MR-PRESSO effect estimates (odds ratios) from our main analyses. Using IVW odds ratios, power was 97% (AMD), 93% (dry AMD), and 86% (wet AMD) for cystatin C, but only 14%, 39%, and 6% for creatinine. Using MR-PRESSO odds ratios, power for cystatin C remained high for AMD (94%) and dry AMD (83%) but was lower for wet AMD (67%). Creatinine analyses remained underpowered (6–30%). These results have been added to the Methods (lines 148-149), Results (lines 228-234), and Discussion (lines 328-335) sections to acknowledge the limited statistical power of our study, particularly for creatinine.

  1. A conclusion figure (graphical abstract) will be very useful for the readers.

→ We appreciate this constructive suggestion. We have created a new conclusion figure summarizing the MR study design, datasets, and main findings in a graphical abstract format. This will help readers grasp the study concept and conclusions more intuitively.

Reviewer 2 Report

Comments and Suggestions for Authors

Thank you very much for inviting me to review the manuscript. The study aims to investigate the potential causal relationship between kidney function markers and AMD and its subtypes using MR. The topic is timely and addresses an area of inconsistent observational evidence. The study is well-designed and clearly written. However, several aspects require clarification and further discussion.

  1. Introduction: Consider adding a sentence that directly states the rationale for focusing on cystatin C over other renal markers, given its potential extra-renal functions.
  2. Discussion: When discussing the inconsistent observational findings, the authors could hypothesize why some studies found null results.
  3. The conclusion is that cystatin C is a causal risk factor for AMD independent of kidney function. However, the evidence is indirect. The authors use serum creatinine as a negative control. But creatinine is a less sensitive marker of early renal impairment compared to cystatin C. The null finding for creatinine does not fully rule out a residual confounding effect of subclinical kidney dysfunction that is captured by the cystatin C instruments.

This manuscript presents a potentially important finding regarding a causal role for cystatin C in AMD pathogenesis, independent of serum creatinine. However, the conclusions need to be tempered due to the discrepancy between MR methods and the indirect evidence for kidney-function independence.

Author Response

Thank you very much for inviting me to review the manuscript. The study aims to investigate the potential causal relationship between kidney function markers and AMD and its subtypes using MR. The topic is timely and addresses an area of inconsistent observational evidence. The study is well-designed and clearly written. However, several aspects require clarification and further discussion.

  1. Introduction: Consider adding a sentence that directly states the rationale for focusing on cystatin C over other renal markers, given its potential extra-renal functions.

→ We thank the reviewer for this important comment. We have revised the Introduction to clarify the rationale for prioritizing cystatin C:

(lines 66-70)

“In addition, cystatin C has been studied for its extra-renal roles, including its involvement in inflammation regulation, and neurological functions [18-20]. In addition, since cystatin C is expressed in ocular tissues [21-23], analyzing its association with AMD pathogenesis would be of significant research value.”

  1. Discussion: When discussing the inconsistent observational findings, the authors could hypothesize why some studies found null results.

→ We thank the reviewer for this important comment. We have added possible explanations in the Discussion section:

(lines 272-277)

“Previous observational studies reporting null associations may have been confounded by residual differences in kidney function, reverse causality, or measurement variability in cystatin C assays. In contrast, MR analyses, by leveraging genetic instruments, are less susceptible to such biases. This study may suggest that the cellular and pathophysiological functions of cystatin C deserve closer investigation beyond its conventional role as a marker of kidney function.”

  1. The conclusion is that cystatin C is a causal risk factor for AMD independent of kidney function. However, the evidence is indirect. The authors use serum creatinine as a negative control. But creatinine is a less sensitive marker of early renal impairment compared to cystatin C. The null finding for creatinine does not fully rule out a residual confounding effect of subclinical kidney dysfunction that is captured by the cystatin C instruments.

→ We appreciate this insightful comment. We have revised the conclusion and Discussion to present a more balanced interpretation:

(lines 308-311 and 340-341)

“The null finding observed with creatinine should be interpreted with caution, as cystatin C is less affected by muscle mass and serves as a more sensitive biomarker for detecting early reductions in eGFR compared with creatinine [51].”

These findings may be partly explained by the greater sensitivity of cystatin C compared with creatinine, highlighting the need ~

  1. This manuscript presents a potentially important finding regarding a causal role for cystatin C in AMD pathogenesis, independent of serum creatinine. However, the conclusions need to be tempered due to the discrepancy between MR methods and the indirect evidence for kidney-function independence.

→ We thank the reviewer for this important comment. We added limitation (lines 328-335).

“Finally, statistical power differed substantially between exposures. For cystatin C, power was high for AMD (97%) and dry AMD (93%), and acceptable for wet AMD based on IVW estimates (86%). However, when MR-PRESSO estimates were used, power for wet AMD was lower (67%). Moreover, creatinine-based analyses had clearly insufficient power (14% for AMD, 39% for dry AMD, and 6% for wet AMD based on IVW; 6–30% with MR-PRESSO). Accordingly, the null findings for creatinine should be interpreted with caution, as they may reflect insufficient power rather than the absence of a true association.”

Reviewer 3 Report

Comments and Suggestions for Authors

INTRO & RRL:

  1. State a precise clinical question specifying AMD subtype, stage, and population denominator.
  2. Please clarify terminology inconsistently mixing early, dry, and wet AMD classification frameworks.
  3. Address selection bias in biobank participation affecting renal markers and ophthalmic outcomes
  4. Articulate translational implications for screening, risk stratification, and therapeutic targeting.

METHODS:

  1. Provide per-SNP F-statistics distribution, not only means, to detect weak instruments.
  2. Conduct colocalization analyses to exclude distinct linked variants driving observed associations.
  3. Specify trait codes, measurement units, and transformations for cystatin C GWAS exposures.
  4. Provide per-SNP F-statistics distribution, not only means, to detect weak instruments.

RESULTS & DISCUSSIONS:

  1. Compare estimates before and after outlier removal, quantifying MR-PRESSO distortion.
  2. Explore nonlinearity using piecewise MR leveraging quantile-specific cystatin C instruments.
  3. Discuss potential horizontal pleiotropy via cathepsin pathway genes

CONCLUSIONS:

1. Provide transparent data harmonization logs enabling independent replication and peer verification.

Author Response

INTRO & RRL:

  1. State a precise clinical question specifying AMD subtype, stage, and population denominator.

→ We appreciate this insightful comment. (lines 33-36 and line 39-44)

“AMD includes dry AMD (early) and wet AMD (late); dry AMD is characterized by drusen accumulation and pigmentary changes [2,3], often without noticeable symptoms; however, it can progress to more severe, wet AMD characterized by two subtypes, i.e., geographic atrophy (GA) and choroidal neovascularization (CNV) that lead to serious vision impairment [4,5].”

“According to a study by Wong et al., the global prevalence of all types of AMD is approximately 8.7% worldwide [7]. In Europe, the prevalence of dry AMD increases from 3.5% in those aged 55–59 years to 17.9% in those aged 85 years; for wet AMD, the prevalence increases from 0.1% to 9.8% [8]. For Asian, the prevalence rates of dry and wet AMD were found to range from 1.4% to 17.3% and 0.1% to 7.3%, respectively, with a higher prevalence in elderly groups [9,10].”

  1. Please clarify terminology inconsistently mixing early, dry, and wet AMD classification frameworks.

→ We appreciate this insightful comment. In accordance with the suggestion, an explanation of the term has been added in lines 33-36. In addition, for consistency throughout the manuscript, the terms “dry AMD” and “wet AMD” were used.

  1. Address selection bias in biobank participation affecting renal markers and ophthalmic outcomes

→ We appreciate the reviewer’s thoughtful comment. We have added a statement in the Discussion section (lines 318-322) acknowledging that both the exposure and outcome GWAS datasets were derived from volunteer-based biobank cohorts, which may introduce selection bias (so-called healthy volunteer bias). Such bias could influence the genetic associations with renal biomarkers and ophthalmic outcomes and may affect the generalizability of our findings to the broader population.

  1. Articulate translational implications for screening, risk stratification, and therapeutic targeting.

→ We appreciate this insightful comment. (line 247-253)

“These findings suggest that cystatin C may serve as a potential biomarker for identifying individuals at increased risk of AMD, beyond its conventional role as a marker of kidney function. Incorporating cystatin C measurements into clinical screening frame-works could improve early detection and refine risk stratification for both dry and wet AMD. Moreover, elucidating the molecular pathways linking cystatin C to retinal de-generation may open new therapeutic avenues, providing opportunities for targeted interventions aimed at modulating cystatin C–related mechanisms in AMD pathogenesis.”

METHODS:

  1. Provide per-SNP F-statistics distribution, not only means, to detect weak instruments.

→ Thank you for your comments. Per-SNP F-statistics are listed in Supplementary Table S1. All of the F statistics for IV MR were greater than 10, indicating a low chance of weak instrument bias.

  1. Conduct colocalization analyses to exclude distinct linked variants driving observed associations.

→ We thank the reviewer for this important suggestion. We conducted locus-specific colocalization across 446 cystatin C instrument regions using the coloc framework (±500 kb; priors p1 = 1 × 10⁻⁴, p2 = 1 × 10⁻⁴, p12 = 1 × 10⁻⁵) after stringent harmonization (see Supplementary Table S3). For AMD, two loci showed strong evidence for a shared causal variant (PP.H4 ≈ 0.99). For dry AMD and wet AMD, one locus each showed strong colocalization (PP.H4 ≈ 0.99). Most regions were PP.H1-dominant, indicating limited local signal, while a subset were PP.H3-dominant (PP.H3 > 0.5), suggesting distinct linked variants. To address this, we repeated MR analyses excluding PP.H3-dominant loci, and the causal estimates remained materially unchanged (see Supplementary Table S4). These results indicate that the main MR conclusions are robust and not driven by LD-linked heterogeneity. Accordingly, we have added this information to the Methods (lines 149-159) and Results sections (lines 235-241).

  1. Specify trait codes, measurement units, and transformations for cystatin C GWAS exposures.

→ We appreciate this helpful comment. The Data Source section has been revised to specify the trait codes, measurement units, and transformation applied to the exposure traits. Specifically, cystatin C (Field ID: 30720, unit: mg/L) and serum creatinine (Field ID: 30700, unit: µmol/L) were both inverse-rank normal transformed before genome-wide association analysis in the Pan-UK Biobank dataset. This information has been added to the revised manuscript.

(lines 101-105)

“In the Pan-UK Biobank, cystatin C (Field ID: 30720, unit: mg/L) and serum creatinine (Field ID: 30700, unit: µmol/L) were measured from serum samples. Both traits were inverse-rank normal transformed prior to genome-wide association analysis to reduce skewness and ensure comparability across participants.”

  1. Provide per-SNP F-statistics distribution, not only means, to detect weak instruments.

→ Thank you for your comments. Per-SNP F-statistics are listed in Supplementary Table S1. All of the F statistics for IV MR were greater than 10, indicating a low chance of weak instrument bias.

RESULTS & DISCUSSIONS:

  1. Compare estimates before and after outlier removal, quantifying MR-PRESSO distortion.

→ We thank the reviewer for this helpful suggestion. To address this comment, we examined whether outlier correction materially affected the causal estimates by performing the MR-PRESSO distortion test for each exposure–outcome pair. The distortion p-values were 0.852, 0.714, and 0.581 for cystatin C on AMD, dry AMD, and wet AMD, respectively, and 0.317, 0.838, and 0.108 for serum creatinine on these same outcomes, indicating no significant distortion after outlier removal. These results confirm that the exclusion of outliers did not meaningfully alter the causal estimates. The corresponding description has been added to the Methods (lines 145-146) and Results sections (lines 218-221 and 226-227) of the revised manuscript.

  1. Explore nonlinearity using piecewise MR leveraging quantile-specific cystatin C instruments.

→ We thank the reviewer for this insightful suggestion. We agree that assessing potential non-linear relationships between cystatin C and AMD could provide additional biological insights. However, because this study was conducted using a two-sample MR framework based entirely on summary-level GWAS data, non-linear (piecewise) MR analyses were not feasible; such methods require individual-level data to construct quantile-specific exposure instruments. We have clarified this methodological limitation in the Discussion (lines 322-328) and noted that future MR analyses based on individual-level data would be valuable to explore potential nonlinearity in these associations.

  1. Discuss potential horizontal pleiotropy via cathepsin pathway genes

We thank the reviewer for highlighting this important mechanistic consideration. We have now added a paragraph in the Discussion section (lines 292-302) discussing the potential for horizontal pleiotropy via cathepsin-pathway genes.

“Given that cystatin C is a potent endogenous inhibitor of lysosomal cathepsins, we considered the possibility of horizontal pleiotropy mediated via cathepsin-axis genes (e.g. CTSB, CTSD, CTSK, CTSS, and CTSV). In this MR study, SNPs associated with rs73102387 (CST3;CST4), the SNPs (rs78516764, rs111701119, rs118011290) encode CST5;GGTLC1 and rs7688550 (CTSO;PDGFC) were used as instrumental variables, which may have influenced the analytical results. Genetic variants in these genes could influence AMD risk through pathways independent of cystatin C and renal function, for instance via extracellular matrix remodeling, angiogenesis, or choroidal neovascu-larisation. Accordingly, while our MR framework mitigates many confounding influences, residual pleiotropic effects via cathepsin-related genes cannot be fully excluded.”

CONCLUSIONS:

  1. Provide transparent data harmonization logs enabling independent replication and peer verification.

→ We thank the reviewer for emphasizing the importance of transparency and reproducibility. In response, we have expanded the Methods section (lines 117-122) to provide a more detailed description of the data harmonization and filtering procedures. In addition, to facilitate independent replication and peer verification, we have made the main analysis scripts—including those used for genetic instrument selection, harmonization, and MR estimation—publicly available on GitHub

“The code for the main analyses in this study is available on GitHub (https://github.com/lyou7688/cystatinC_AMD_MR).”

We have included this statement in the Data Availability section.

Round 2

Reviewer 1 Report

Comments and Suggestions for Authors

well revision

Reviewer 2 Report

Comments and Suggestions for Authors

I have no further comments, congratulations to all authors.

Reviewer 3 Report

Comments and Suggestions for Authors

My comments have been address on the revised version. Proofread the work.